# SARS-CoV-2 RNA in Wastewater and Bivalve Mollusk Samples of Campania, Southern Italy

**DOI:** 10.3390/v15081777

**Published:** 2023-08-21

**Authors:** Annalisa Lombardi, Antonia Voli, Andrea Mancusi, Santa Girardi, Yolande Thérèse Rose Proroga, Biancamaria Pierri, Renato Olivares, Luigi Cossentino, Elisabetta Suffredini, Giuseppina La Rosa, Giovanna Fusco, Antonio Pizzolante, Amalia Porta, Pietro Campiglia, Ida Torre, Francesca Pennino, Alessandra Tosco

**Affiliations:** 1Department of Public Health, University “Federico II”, Via Sergio Pansini 5, 80131 Naples, Italy; annalisa.lombardi@unina.it (A.L.);; 2Department of Pharmacy, University of Salerno, Via Giovanni Paolo II 132, 84084 Fisciano, Italy; avoli@unisa.it (A.V.); aporta@unisa.it (A.P.); pcampiglia@unisa.it (P.C.); 3Department of Food Security Coordination, Zooprophylactic Institute of Southern Italy, Via Salute 2, 80055 Portici, Italy; andrea.mancusi@izsmportici.it (A.M.); santa.girardi@izsmportici.it (S.G.); yolande.proroga@izsmportici.it (Y.T.R.P.); biancamaria.pierri@izsmportici.it (B.P.); 4Campania Regional Environmental Protection Agency (ARPAC), Via Vicinale Santa Maria del Pianto, 80143 Naples, Italy; r.olivares@arpacampania.it (R.O.); l.cossentino@arpacampania.it (L.C.); 5Department of Food Safety, Nutrition and Veterinary Public Health, Istituto Superiore di Sanità, Viale Regina Elena 299, 00161 Rome, Italy; elisabetta.suffredini@iss.it; 6Department of Environment and Health, Istituto Superiore di Sanità, Viale Regina Elena 299, 00161 Rome, Italy; giuseppina.larosa@iss.it; 7Zooprophylactic Institute of Southern Italy, Via Salute 2, 80055 Portici, Italy; giovanna.fusco@izsmportici.it (G.F.); antonio.pizzolante@izsmportici.it (A.P.)

**Keywords:** SARS-CoV-2, wastewater, bivalve mollusk, RNA, wastewater treatment plants, epidemiological data

## Abstract

SARS-CoV-2 can be detected in the feces of infected people, consequently in wastewater, and in bivalve mollusks, that are able to accumulate viruses due to their ability to filter large amounts of water. This study aimed to monitor SARS-CoV-2 RNA presence in 168 raw wastewater samples collected from six wastewater treatment plants (WWTPs) and 57 mollusk samples obtained from eight harvesting sites in Campania, Italy. The monitoring period spanned from October 2021 to April 2022, and the results were compared and correlated with the epidemiological situation. In sewage, the ORF1b region of SARS-CoV-2 was detected using RT-qPCR, while in mollusks, three targets—RdRp, ORF1b, and E—were identified via RT-dPCR. Results showed a 92.3% rate of positive wastewater samples with increased genomic copies (g.c.)/(day*inhabitant) in December–January and March–April 2022. In the entire observation period, 54.4% of mollusks tested positive for at least one SARS-CoV-2 target, and the rate of positive samples showed a trend similar to that of the wastewater samples. The lower SARS-CoV-2 positivity rate in bivalve mollusks compared to sewages is a direct consequence of the seawater dilution effect. Our data confirm that both sample types can be used as sentinels to detect SARS-CoV-2 in the environment and suggest their potential use in obtaining complementary information on SARS-CoV-2.

## 1. Introduction

SARS-CoV-2 is a single-stranded RNA virus belonging to the *Coronaviridae* family responsible for the COVID-19 (Coronavirus disease 2019) pandemic that emerged in 2020 [1]; from December 2019 to June 2023, about 770 million people globally were infected with SARS-CoV-2, of which almost 7 million died [2].

The rapid spread of COVID-19 and the high mortality rate have revealed the urgent need for robust monitoring and surveillance methods for this and other potential pathogens. SARS-CoV-2 has been detected in the feces of infected subjects and, consequently, in sewage [3]; for this reason, Wastewater-Based Epidemiology (WBE) has been indicated as a valuable tool that can provide information on human infection trends and early indications of future outbreaks [4].

In July 2020, the Italian National Institute of Health (Istituto Superiore di Sanità, ISS) launched a pilot project named SARI (Epidemiological Surveillance for SARS-CoV-2 in Urban Sewage) to investigate SARS-CoV-2 in wastewater. Soon after, the EU Commission Recommendation 2021/472, issued on 17 March 2021, encouraged member states to establish a surveillance system for SARS-CoV-2 and its variants no later than 1st October 2021. The Recommendation also provided guidance on sampling, testing methods, and specific quality standards to obtain comparable results (Recommendation 2021/472). In agreement with this document, the ISS converted the pilot project into a surveillance system and implemented it for monitoring SARS-CoV-2 in wastewater on a national scale [5].

Once treated in the wastewater implants, virus residues reach the sea, even if in a diluted concentration, and it is well known that they can also be found in bivalve mollusks, including oysters, mussels, and clams since these aquatic animals are able to accumulate viruses due to their ability to filter large amounts of water (each kg of mussel can filter between 100 and 650 L of water in one hour) [6]. Indeed, bivalve mollusks are considered a vector of enteric viruses, especially if consumed raw or not properly cooked; the treatments they are subjected to after collection do not always allow the total elimination of viruses. However, the probability of transmission of SARS-CoV-2 from ingesting these organisms seems extremely low [7].

Some studies have detected SARS-CoV-2 RNA in bivalve mollusks. Le Guernic et al. [8] tested the presence of the virus in zebra mussel samples exposed to treated and untreated wastewater, and Polo et al. [7] described the presence of SARS-CoV-2 RNA traces in clam samples and marine sediments. Moreover, Mancusi et al. recently identified SARS-CoV-2 RNA in mollusks samples collected in Campania, Southern Italy, from September 2019 to April 2021 and observed that a high percentage of positive mollusk samples corresponded to periods of increased circulation of SARS-CoV-2 in the population [9]. To our knowledge, only two studies investigated the presence of SARS-CoV-2 RNA in both wastewaters and bivalve mollusks in Croatia [10] and Spain [11], respectively.

This study aimed to investigate the occurrence of SARS-CoV-2 RNA in both wastewater and bivalve mollusks samples collected in the Campania region from October 2021 to April 2022. Specifically, the aims were (i) to monitor the presence of SARS-CoV-2 RNA in wastewater treatment plants in Campania; (ii) to monitor the presence of SARS-CoV-2 RNA in bivalve mollusks harvested along the coasts of the same region; (iii) to assess whether there was a temporal correspondence between the results obtained in wastewater and mollusk samples, in order to evaluate the feasibility of using both sample types as sentinels for monitoring the epidemiological situation in a geographic area, supporting public health surveillance.

## 2. Materials and Methods

### 2.1. Wastewater Treatment Plants and Wastewater Sampling

Raw wastewater samples (N = 168) were collected once a week from October 2021 to April 2022 at the inlet of 6 wastewater treatment plants (WWTPs) located in Campania (Southern Italy). Specifically, samples were derived from the following WWTPs: Napoli EST, Napoli OVEST (two inlets, identified as ‘Main inlet’ and ‘North sewer’), Area Nolana, Area Casertana, and Nocera Superiore. The characteristics of these six WWTPs, including their final discharge point, are reported in Table 1, and their geographical coordinates are shown in Figure 1. 

The sampling and analysis of these wastewater samples were undertaken as a part of the national SARS-CoV-2 surveillance program in wastewater (SARI project), and the WWTPs were selected for being representative of Campania’s population. Samples were collected, processed and analyzed for SARS-CoV-2 identification according to the SARI protocol, revision 3 [12]. For each WWTP, 500 mL of a 24 h composite sample were collected into polyethylene bottles with an automatic sampler and transported in thermal bags at a temperature of 4 °C to the laboratories, where they underwent a virus inactivation step at 56 °C for 30 min to ensure the safety of the laboratory personnel and the environment during the analytical procedure. The samples were analyzed within 24 h of collection.

### 2.2. Wastewater Processing and RNA Extraction

Wastewater analysis was performed using a PEG precipitation protocol [13], as detailed in the SARI protocol [12]. Before concentration, 100 µL of a process control virus (Mengovirus or Murine Norovirus), supplied by ISS, was added to each 45 mL sample to monitor the efficiency of the concentration and extraction process. Viral RNA extraction was performed using the NucliSENS kit with magnetic silica, following the manufacturer’s instructions (bioMerieux, Marcy-l’Étoile, France). In detail, the PEG precipitate was directly resuspended with 2 mL of lysis buffer and incubated for 20 min at room temperature, 50 µL of magnetic silica beads were added, and after an incubation of 10 min and two washing steps, the elution was performed with 100 µL of the buffer included in the kit. Before molecular analysis, the nucleic acids were further purified from potential PCR inhibitors using OneStep PCR Inhibitor Removal Kit (Zymo Research, Irvine, CA, USA).

### 2.3. Real-Time RT-PCR for SARS-CoV-2 Detection in Wastewater Samples

Real-time RT-PCR (RT-qPCR) was performed to identify the ORF1b (nsp14; 3′-to-5′ exonuclease) of SARS-CoV-2 [14]. The thermal profile consisted of an initial step at 50 °C for 30 min, followed by a denaturation step at 95 °C for 10 min, and 45 cycles of 95 °C for 15 s and 60 °C for 45 s. The primers and probes sequences are shown in Table 2. Primers 2297-CoV-F and 2298-CoV-R and probe 2299-CoV-P were used for SARS-CoV-2 detection in wastewater. All reactions were performed in technical duplicate. The reaction mix contained 20 µL of AgPath-ID One-Step RT-PCR Reagents (Applied Biosystems, Waltham, MA, USA) and 5 µL of each sample. The results were expressed as genomic copies (g.c.)/µL of RNA by interpolating the cycle threshold (Ct) values with a standard curve obtained through serial dilutions of the dsDNA of SARS-CoV-2 nsp14 provided by ISS.

An external inhibition control (an in vitro synthesized RNA containing the target sequence supplied by ISS) was used to check for the presence of PCR inhibitors. The sample was considered acceptable if the ΔCt between the sample and the reference (uninhibited reaction containing the target in molecular-grade water) was ≤2. 

For monitoring the efficiency of the concentration/extraction process, the Ct values of the process control virus recovered from the samples were compared with serial dilutions of the virus used for spiking. The samples were considered acceptable if the concentration/extraction efficiency was ≥1%.

All calculations related to SARS-CoV-2 quantification, inhibition control and recovery were performed with the validated Excel calculation file distributed by ISS to the laboratories participating in the SARI project.

### 2.4. Study Area and Bivalve Mollusk Sampling

Bivalve mollusks analysis was carried out as described in Mancusi et al. [9]. A total of 57 bivalve mollusk samples were collected from various locations, as indicated in Figure 2. The sampling sites and samples were as follows: 16 originate from Bacoli (Lago Fusaro, Punta Cento Camerelle—Punta del Poggio, and Punta Terone—Capo Miseno), 4 from Castellammare di Stabia—Molo Foraneo, 4 from Ercolano, 8 from Giugliano in Campania—Varcaturo, 7 from Monte di Procida—Acquamorta, 6 from Napoli—Nisida—Punta Cavallo, 6 from Napoli—Rada Santa Lucia Est and Ovest, 6 from Torre Annunziata—Punta Oncino. Samples were collected during official surveillance plans implemented by the local health services to prevent and detect illegal harvesting practices. The sampling frequency was variable and ranged from no sampling to a maximum of 4 times per month. The sampling frequency variability was due to weather conditions (e.g., wind or turbulent sea) that influenced the collection of bivalve mollusks samples. Upon collection, the samples were transported to the laboratory at 4 °C and processed within 24 h.

### 2.5. Bivalve Mollusk Samples Preparation and SARS-CoV-2 RNA Extraction

SARS-CoV-2 RNA recovery from bivalve mollusk samples was performed according to ISO 15216-2:2019. To verify extraction efficiency, 10 µL of Mengovirus (process control virus) were added to 2 g of digestive glands (hepatopancreas) homogenate. Tissue digestion was obtained by treating the homogenate with 2 mL of a proteinase K solution (Qiagen, Hilden, Germany). The mixture was then incubated at 37 °C for 60 min with shaking at 320 rpm and 60 °C for 15 min. Subsequently, tubes were centrifuged at 3000× *g* for 5 min, and the supernatant was collected in another tube, measured, and analyzed. RNA extraction was obtained from 500 µL of supernatant using the NucliSENS reagents (bioMerieux), following the manufacturer’s instruction, with the final elution in 100 µL. 

### 2.6. Droplet Digital RT-PCR (RT-dPCR) for SARS-CoV-2 Detection in Mollusks

Two specific targets, the RdRp gene [15] and the orf1b gene, nsp14 [14,18], and a generic target for beta-Coronavirus, the E gene [15], were used to detect SARS-CoV-2. The QX200 system (Bio-RAD, Hercules, CA, USA) was used to perform dd RT-PCR. The reaction was performed in a volume of 20 µL, including 1X One-step RT-dPCR Advanced Kit for Probes, 20 U/µL reverse transcriptase, 15 mM DTT, 5 µL of RNA sample, and nuclease-free water as required. Sequences of primers and probes used for RdRp, E gene, and ORF1ab nsp14 are reported in Table 2.

The reaction mixtures were placed in DG8 cartridge wells (Bio-Rad Laboratories, California, USA) with 70 µL of droplet generation oil, and droplets were formed in the droplet generator (Bio-Rad Laboratories, California, USA). Then, 40 µL of droplet-partitioned samples were transferred to a 96-well plate, sealed, and amplified on a CFX96 Deep Well instrument (Bio-Rad Laboratories, California, USA) with the following thermal profile: 50 °C for 60 min, 95 °C for 10 min followed by 45 cycles of 95 °C for 15 s and 60 °C for 45 s, and a final stage at 98 °C for 10 min. At the end of the amplification cycles, the 96-well plate was read in the QX200 Droplet Reader, which detects positive droplets according to the Poisson distribution. QuantSoft software version 1.7 was used to count the PCR-positive and PCR-negative droplets to provide absolute quantification of target DNA. The quantification of each target was expressed as the number of g.c./µL of the RNA sample. Positive controls were included, consisting of SARS-CoV-2 RNA containing the target genes (E, RdRp and ORF1b), certified by Bio-Rad Laboratories (Bio-Rad). The assay’s Limit of Detection (LOD) was 1 g.c./µL of RNA for each target. 

### 2.7. Data Analysis

The results of wastewater samples were expressed in terms of g.c./µL of RNA and g.c./L of sewage (using a conversion formula). The latter was further multiplied by the plant flow rate (expressed in m^3^/day) and then normalized with the number of equivalent inhabitants. On the other hand, the results of the bivalve mollusk samples were reported as g.c./µL of RNA, then converted to g.c./g of digestive tissue of bivalve mollusk. To accommodate the different sampling frequencies, weekly and monthly values were calculated for the wastewater and the mollusks samples, respectively. 

### 2.8. Statistical Analysis

Statistical analysis was descriptive. The mean, minimum and maximum values were calculated using Microsoft Excel.

## 3. Results

### 3.1. Results of Wastewater Monitoring

A total of 155 wastewater samples out of 168 (92.3%) tested positive for SARS-CoV-2 detected using RT-qPCR. The viral concentration of positive samples ranged from a minimum of 1.73 × 10^2^ g.c./L to a maximum of 2.90 × 10^5^ (corresponding to Napoli OVEST—North sewer WWTP on 7 April 2022) g.c./L, and from 2.11 × 10^4^ g.c./(day*inhabitant) (corresponding to Napoli EST WWTP of 28 October 2021) to 5.76 × 10^7^ g.c./(day*inhabitant) (corresponding to Area Casertana WWTP of 7 April 2022). Figure 3 illustrates the temporal trends of g.c./(day*inhabitant) values showing that there was an initial increase in late December 2021–January 2022, except for the Area Casertana WWTP, followed by a second increase in March 2022–April 2022. Note that the concentrations registered in the second peak were higher than those during the first. 

The highest percentage of negative values was observed in the Area Casertana WWTP (28.6%, 8 positive samples out of 28 collected samples); this WWTP is also the only one that did not record the first increase in concentrations from late December 2021 to January 2022.

### 3.2. Results of Bivalve Mollusks Monitoring

Thirty-one of the fifty-seven bivalve mollusk samples (54.4%) were positive for at least one of the SARS-CoV-2 targets analyzed via RT-dPCR (Table 3. In detail, 3 samples (9.7%) tested positive for all three tested targets (RdRp, ORF1b-nsp14 and E gene), 10 (32.2%) for two different targets (RdRp and ORF1b), 7 (22.6%) either only for the RdRp gene or for ORF1b, and 4 (12.9%) only for E gene. 

Depending on the PCR target, the viral concentration ranged from 8.4 × 10^1^ and 1.1 × 10^3^ g.c./g of digestive tissue for the RdRp gene analysis, from 7.8 × 10^1^ to 2.0 × 10^3^ g.c./g for the orf1b-nsp14 target, and from 7.8 × 10^1^ to 4.8 × 10^2^ for the E gene.

As shown in Table 4, the percentages of positive samples in each sampling site (considering all the samples with at least one positive gene) were consistently above 40% for the entire period, suggesting a high virus circulation in the population during the considered months.

### 3.3. Comparison of Wastewater and Mollusk Samples Results

Figure 4 shows the average viral load in the inlet of all monitored WWTPs, the percentage of positive bivalve mollusk samples of the monitored area, and the number of COVID-19 cases in the population based on daily swab tests.

The average of the viral load in sewage samples of the Campania region shows an initial increase in December 2021, followed by a peak in January 2022 and a second increase in March 2022, even more pronounced in April 2022 (Figure 4A). In particular, from December to March, it a progressive bioaccumulation of the virus over time (Figure 4B) was observed. Finally, if we consider the active cases in Campania in this period, we observe a considerable increase towards the end of December 2021, followed by a high peak in January 2022 and another rise in March–April (Figure 4C). According to the high spreading rate of SARS-CoV-2 in the population during the considered period, we obtained a high percentage of positives both in wastewater and in mollusks samples and peaks of viral load in sewage corresponding to those obtained from swab tests on the inhabitants of Campania.

## 4. Discussion

SARS-CoV-2 originated in China in December 2019; within a few months it spread worldwide and was declared an international public health problem in January 2020 and a pandemic in March of the same year [19].

This study aimed to monitor the presence of SARS-CoV-2 RNA in both wastewater and bivalve mollusk samples in Campania to explore the use of these samples as sentinels of an epidemiological situation in a specific area. We used RT-qPCR to identify SARS-CoV-2 in wastewater and used RT-dPCR in mollusks. The first is a widely used and versatile method that requires 1–2 h for sample analysis but is susceptible to the presence of PCR inhibitors and requires generating a standard curve to provide quantitative results. In contrast, digital PCR does not require such a curve, is less affected by PCR inhibitors, is more analytically sensitive and can identify mutations. On the other hand, it is a more expensive method than RT-qPCR, requiring more care in loading samples and more time for preparation and analysis [20]. We used RT-qPCR for wastewater samples following the recommendations of the SARI protocol, revision 3; nevertheless, we decided to use RT-dPCR for mollusks analysis for its higher sensitivity to reveal the low viral levels of virus in the tissues of these organisms [9].

Surveillance of SARS-CoV-2 in wastewater has emerged as a valuable tool for monitoring epidemic trends and protecting public health [21]. As reported by several studies, wastewater surveillance enables the early detection of infections in the population, often preceding clinical data [13,22,23,24]; Smith et al. [25] also endorsed its role in supporting vaccination efforts, revealing areas or communities with increased viral transmission or new outbreaks. In addition, wastewater analysis provides insights into infectious trends in the population regardless of the availability of clinical testing data [21]. Environmental monitoring of SARS-CoV-2 has been extensively described also in various Italian regions [22,26,27,28,29,30,31,32], confirming it as a powerful tool for tracking the spread of SARS-CoV-2 variants/subvariants in the population [32,33,34].

On the other hand, monitoring SARS-CoV-2 in bivalve mollusks provides us with an opportunity to explore the potential use of these shellfish as sentinel organisms [35] since the ability of bivalve mollusks to bioaccumulate the SARS-CoV-2 genome has already been demonstrated. Le Guernic et al. performed experiments exposing zebra mussels (*Dreissena polymorpha*) to treated and untreated wastewater from French WWTPs and revealed the presence of the SARS-CoV-2 genome in their digestive tissues in both conditions, analyzing E, RdRp, and N genes. These results highlighted the persistence of the virus genome even in the treated water and simultaneously confirmed the possibility of using mussels as sentinels [8]. Another study carried out in Galicia, Spain, from May to July 2020 revealed the presence of the SARS-CoV-2 genome in 9 clam samples (*Ruditapes* sp.) out of 12 tested; even in this case, the analyzed genes were N1, RdRp, and E. Among the positive samples, the simultaneous presence of two positive genes was revealed in four of them, and the viral load in the mollusk tissues ranged from a minimum of 0 g.c./g to a maximum of 4.48 log [7]. Novoa et al. also monitored the presence of SARS-CoV-2 in sewage and mussels in Galicia, Spain, and observed a decrease in viral load in treated wastewater and, consequently, in mussels [11]. 

Moreover, the study by Tiwari et al. assessed the virus’s viability in Finnish wastewater samples using cell cultures, and the results showed the absence of infectious particles in the analyzed samples, suggesting that the water route does not represent a way for virus transmission [36]. However, a study by Ahmed et al. revealed little chance of the transmission of SARS-CoV-2 from the water route, indicating the generation of fecal aerosols from such water as the most likely mechanism of infection [37]. However, the detection of the SARS-CoV-2 genome in bivalve mollusks could also be an indicator of several phenomena, such as the uncontrolled release of sewage into the sea, the efficacy of wastewater treatment methods in removing viruses, or the different pathways that wastewater may take without passing through wastewater treatment plants [11].

Different environmental monitoring investigations have been piloted on the wastewaters of Campania in recent years [35,38,39,40,41,42,43]. In our study, we collected samples from harvesting areas of Campania and observed that a high percentage of positive mollusk samples corresponded to periods of increased circulation of SARS-CoV-2 in the population [9]. In the period under investigation, from October 2021 to April 2022, we detected SARS-CoV-2 viral loads with values ranging from 10^2^ to 10^5^ g.c./L in wastewater. Note that a comprehensive view of the presence of SARS-CoV-2 RNA in Italian wastewater in November–December 2021 can be found in La Rosa et al., where g.c./L values between 10^2^ and 10^6^ are reported [33].

All WWTPs analyzed in our study showed a similar trend of SARS-CoV-2 viral load over time, with a first increase in late December 2021–January 2022 and a second increase in March–April 2022. This finding suggests that the trend of SARS-CoV-2 in the sewage, and consequently in the population, was similar among different areas. The only exception was the Area Casertana WWTP, whose high rate of negative occurrence could be explained by its proximity to an industrial site, which could have diluted the sample and decreased the RNA viral load.

Regarding the mollusk samples, we analyzed three different genes, observing a trend of bioaccumulation similar to that seen in wastewater, with a high percentage of positive samples from December to April. However, our results show a higher rate of positive samples among sewages than among mollusks out of the total number of samples analyzed. This agrees with the study of Novoa et al. [11], who observed a lower impact of SARS-CoV-2 in wild and aquaculture mussels. Viral RNA load is already diluted in wastewater and becomes even more diluted in the seawater, reaching levels below the limit of detection; furthermore, the signal is also affected by the sewage treatment before reaching the final discharge. As a result, the bioaccumulation in the mussel digestive tract can be appreciated mainly when the spread in the population is high or when failures in the treatment system occur.

In addition, the presence of SARS-CoV-2 RNA in bivalve mollusks is an indicator of the presence of its genome in the effluent wastewaters [11], highlighting an incomplete removal by the treatments to which the inlets are subjected, as also observed in the studies by Battistone et al. [38] and Pennino et al. [43] investigating Enterovirus; our results suggest the need to search treated effluents for viral parameters. 

The field of “sewage science,” as reported by Diamond et al. [44], gained recognition starting during the polio eradication campaign [45], and its importance only increased throughout the current COVID-19 pandemic [46]. The potential for wastewater monitoring as a multi-pathogen tool for epidemic surveillance is undeniable, particularly in the context of emerging and re-emerging threats to human health [44]. One potential improvement for wastewater monitoring lies in the simultaneous monitoring of sentinel organisms, such as mollusks. This approach could complement the detection of pathogens in sewage, implementing an integrated model for public health surveillance.

## 5. Limitations

This study has several limitations that should be acknowledged. One of the main challenges encountered was the inability to sample bivalve mollusks at a steady and high frequency, similar to the sampling frequency employed for sewage samples. It would be worthwhile to compare the weekly trends of SARS-CoV-2 in both types of samples (e.g., by establishing dedicated bivalve mollusks installations for biomonitoring purposes) to assess any possible correlations. Future research could address this knowledge gap and provide valuable insights, improving the integrated model of public health surveillance.

## 6. Conclusions

The simultaneous collection of sewage and mussel samples and their observation data show the state of viral circulation in the population and provide an adequate representation of the infection trend without the need to invest new economic resources in large-scale swabs. The results of this study confirm that environmental surveillance has the potential to document the diffusion of the virus and suggest the use of these samples for monitoring purposes.

## Figures and Tables

**Figure 1 viruses-15-01777-f001:**
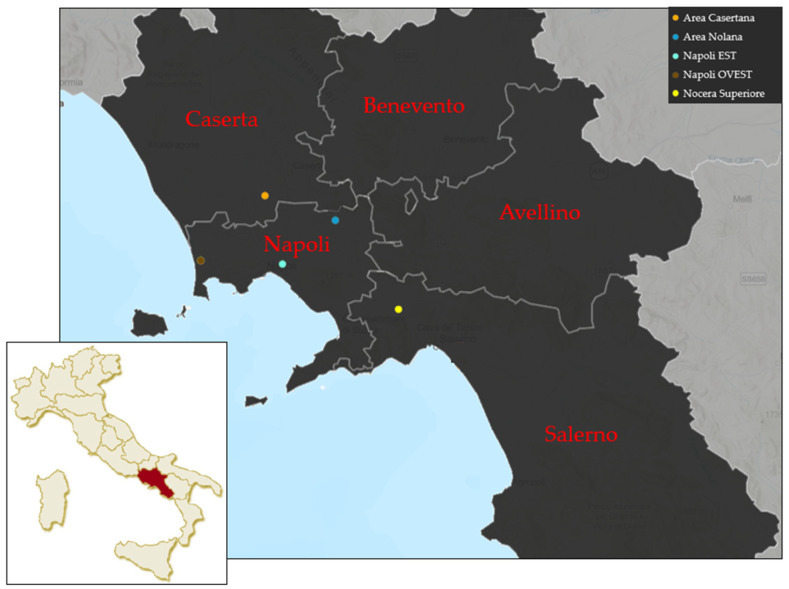
Wastewater sampling area map.

**Figure 2 viruses-15-01777-f002:**
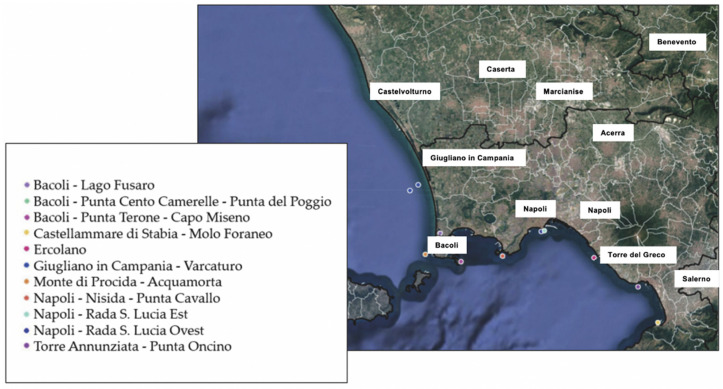
Bivalve mollusks sampling area map.

**Figure 3 viruses-15-01777-f003:**
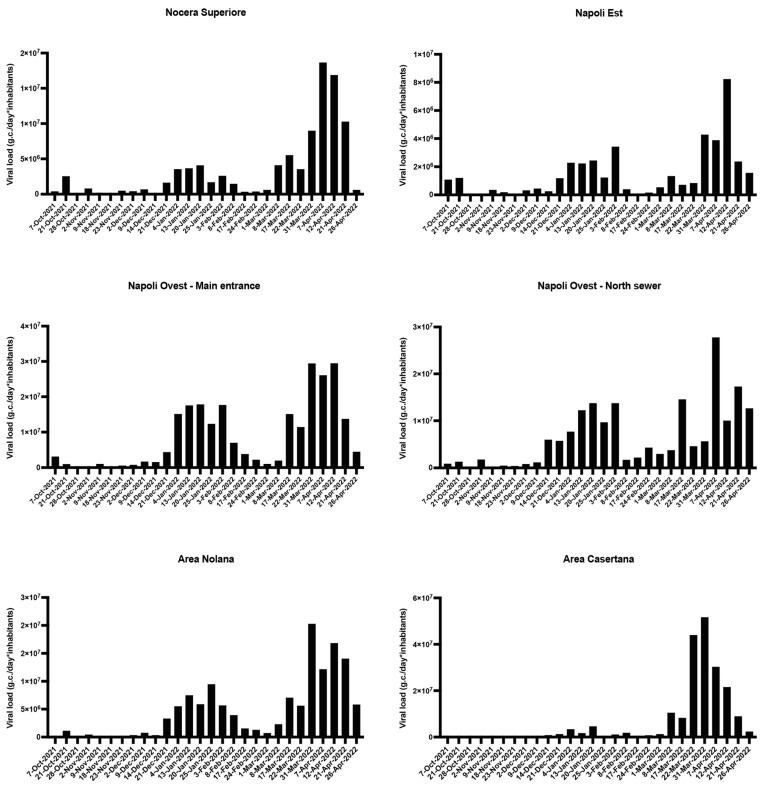
SARS-CoV-2 load in monitored WWTPs between October 2021 and April 2022.

**Figure 4 viruses-15-01777-f004:**
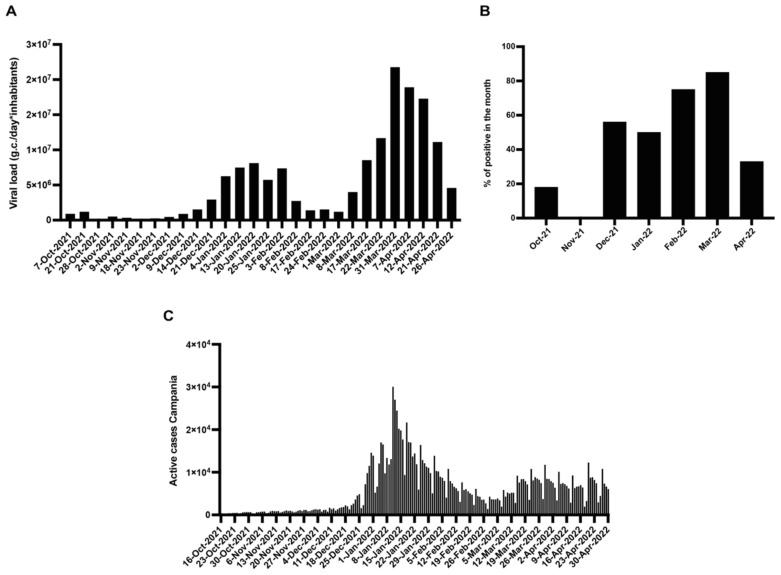
Weekly viral load average of all monitored WWTPs (**A**); monthly percentage of positive bivalve mollusk samples of all the sampling sites (**B**); daily number of COVID-19 cases in Campania (**C**).

**Table 1 viruses-15-01777-t001:** WWTPs monitored and equivalent inhabitants.

WWTP	Equivalent Inhabitants	Depurative Process	Final Receptor	Number of Municipalities Served
NAPOLI EST	1,750,000	Chemical-physical	Tyrrhenian Sea	11
NAPOLI OVEST—Main entrance	950,000	Biologic-Active Sludge	Tyrrhenian Sea	2
NAPOLI OVEST—North sewer	250,000	Biologic-Active Sludge	Tyrrhenian Sea	7
AREA NOLANA	400,000	Biologic-Active Sludge	Regi Lagni canals	35
AREA CASERTANA	370,769	Biologic-Active Sludge	Regi Lagni canals	16
NOCERA SUPERIORE	299,121	Biologic-Active Sludge	Sarno River	5

**Table 2 viruses-15-01777-t002:** Oligonucleotide name, sequence and reference used in this study.

Oligonucleotide	Sequence	Reference
RdRP_SARSr-F	GTG ARA TGG TCAT GTG TGG CGG	[15]
RdRP_SARSr-R	CAR ATG TTA AAS ACA CTA TTA GCAT A	[15]
RdRP_SARSr-P1	FAM-CCA GGT GGW ACR TCA TCM GGT GAT GC-BBQ	[15]
RdRP_SARSr-P2	FAM-CAG GTG GAA CCT CAT CAG GAG ATG C-BBQ	[15]
E_Sarbeco_F	ACA GGT ACG TTA ATA GTT AAT AGC GT	[15]
E_Sarbeco_R	ATA TTG CAG CAG TAC GCA CAC A	[15]
E_Sarbeco_P1	FAM-ACA CTA GCC ATC CTT ACT GCG CTT CG-BBQ	[15]
2297-CoV-F	ACA TGG CTT TGA GTT GAC ATC T	[14]
2298-CoV-R	AGC AGT GGA AAA GCA TGT GG	[14]
2299-CoV-P	FAM-CAT AGA CAA CAG GTG CGC TC-MGBEQ	[14]
Mengo 110 (FW)	GCG GGT CCT GCC GAA AGT	[16]
Mengo 209 (REV)	GAA GTA ACA TAT AGA CAG ACG CAC AC	[16]
Mengo 147 (PROBE)	FAM-ATC ACA TTA CTG GCC GAA GC-MGBNFQ	[16]
MNV orf1/2junct (FW)	CAG GCC ACC GAT CTG TTC TG	[17]
MNV orf1/2junct (REV)	GCG CTG CGC CAT TC	[17]
MNV orf1/2junct (PROBE)	FAM-CGC TTT GGA ACA ATG-MGBNFQ	[17]

FAM: 6-Carboxyfluorescein; BBQ: blackberry quencher; MGBEQ: Minor Groove Binder Eclipse Quencher; MGBEQ: Minor Groove Binder Non-Fluorescent Quencher.

**Table 3 viruses-15-01777-t003:** Detection of SARS-CoV-2 genes in bivalve mollusks samples.

	Number of Samples	RdRp Gene	ORF1b, nsp14	E Gene
	3	+	+	+
	10	+	+	-
	7	+	-	-
	7	-	+	-
	4	-	-	+
	26	-	-	-
Total	57	20	20	7

**Table 4 viruses-15-01777-t004:** Detection of SARS-CoV-2 in bivalve mollusk samples according to the sampling site.

Sampling Site	Position in the Coastline	SARS-CoV-2 Detection	
		N° of Positive Samples/N° of Sampling	% of Positive Samples in the Site
Varcaturo	North	5/8	62
Bacoli	North	6/16	37
Monte di Procida	North	3/7	43
Nisida	North	5/6	83
Rada Santa Lucia	Center	3/6	50
Castellammare di Stabia	South	3/4	75
Ercolano	South	3/4	75
Torre Annunziata	South	3/6	50

## Data Availability

Not applicable.

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
