# Peer review of "SARS-CoV-2 RNA in Wastewater and Bivalve Mollusk Samples of Campania, Southern Italy"

_viruses, 2023, doi:10.3390/v15081777_

Round 1

Reviewer 1 Report

The authors investigated the concentration of SARS-CoV-2 in wastewater and in bivalve mollusk samples. The quantity and the quality of the results are enough for the publication. The reviewer suggests minor revision before the publication to the journal.

1) In the section 2.3 “Real-Time RT-PCR for SARS-CoV-2 detection in wastewater samples”, the names of the primers (2297-CoV-F, 2298-CoV-R) and the probe (2299-CoV-P) are advised to be included in the main text, because table 2 include several sets of primers.

2) In the section 2.2 “Wastewater processing and RNA extraction”, the volume of wastewater, the volume of PEG precipitate, and the final volume after the extraction steps should be mentioned, although these volumes may partly be shown in [12,13].

3) In the section “Droplet Digital RT-PCR (dd RT-PCR) for SARS-CoV-2 detection in mollusks”, two probe sequences are given for the RdRp gene. Please mention which of these two probes was used in the detection, although it may be mentioned in [15].

4) The authors can merge paragraphs (line 40 – 52, line 313-322) to form one paragraph because too short paragraph consisting of only one or two sentences is not appropriate.

5) The authors can discuss (or briefly mention) the reason why the authors used real-time PCR for wastewater samples and droplet digital PCR for mollusks samples.

Reviewer 2 Report

Annalisa Lombardi reported the monitoring of SARS-CoV-2 RNA in wastewater and mollusk samples in Italy. Overall they have conducted the study well and presented it well. But still, the presentation could have been better organized. Instead of one sentence paragraph can have a sizable paragraph. It would be talked about the public health risk of detection of SARS-CoV-2 in mollusk samples. The detection of RNA or the detection of active virome particles. However, earlier many studies (e.g. https://doi.org/10.1016/j.watres.2022.118220) failed to get viable virus particles from wastewater. The next study (https://doi.org/10.1093/femsmc/xtab007) also reviewed little chance of the transmission of SARS-CoV-2 from the water route. As, used RT-qPCR/RT-dPCR methods, better to touch on some performance of these two platforms in discussion (e.g. https://doi.org/10.1016/j.scitotenv.2022.155663)

Line 23- how it is in bivalve mollusks can you elaborate?

Line- 27 and 28 Do You mean RT-qPCR/RT-dPCR?

Line 49- What does this subject mean?

Lines-63-74- how does SARS RNA reach to sea can you explain it  first

Line 97- please can you cite this reference properly

Line 233- Do you mean SARS-CoV-2 assay?

I think the paper can be sent to a professional English language correction before the next submission. 
